# Recent Optical Coherence Tomography (OCT) Innovations for Increased Accessibility and Remote Surveillance

**DOI:** 10.3390/bioengineering12050441

**Published:** 2025-04-23

**Authors:** Brigid C. Devine, Alan B. Dogan, Warren M. Sobol

**Affiliations:** 1College of Medicine, Northeast Ohio Medical University, Rootstown, OH 44272, USA; 2Virginia Tech Carilion School of Medicine, Roanoke, VA 24016, USA; 3Department of Ophthalmology and Visual Sciences, Case Western Reserve University School of Medicine/University Hospitals Cleveland Medical Center, Cleveland, OH 44106, USA; warren.sobol@uhhospitals.org; 4Louis Stokes Cleveland Department of Veterans Affairs Medical Center, Cleveland, OH 44106, USA

**Keywords:** retinal telemedicine, self-administered OCT, remote data transfer

## Abstract

Optical Coherence Tomography (OCT) has revolutionized the diagnosis and management of retinal diseases, offering high-resolution, cross-sectional imaging that aids in early detection and continuous monitoring. However, traditional OCT devices are limited to clinical settings and require a technician to operate, which poses accessibility challenges such as a lack of appointment availability, patient and family burden of frequent transportation, and heightened healthcare costs, especially when treatable pathology is undetected. With the increasing global burden of retinal conditions such as age-related macular degeneration (AMD) and diabetic retinopathy, there is a critical need for improved accessibility in the detection of retinal diseases. Advances in biomedical engineering have led to innovations such as portable models, community-based systems, and artificial intelligence-enabled image analysis. The SightSync OCT is a community-based, technician-free device designed to enhance accessibility while ensuring secure data transfer and high-quality imaging (6 × 6 mm resolution, 80,000 A-scans/s). With its compact design and potential for remote interpretation, SightSync widens the possibility for community-based screening for vision-threatening retinal diseases. By integrating innovations in OCT imaging, the future of monitoring for retinal disease can be transformed to reduce barriers to care and improve patient outcomes. This article discusses the evolution of OCT technology, its role in the diagnosis and management of retinal diseases, and how novel engineering solutions like SightSync OCT are transforming accessibility in retinal imaging.

## 1. Introduction

Retinal disease is a leading cause of vision impairment globally [1,2]. Common diseases such as age-related macular degeneration (AMD), which impairs central vision, and diabetic retinopathy, which damages blood vessels in the retina, are growing in prevalence among an aging population and increased rates of diabetes and other chronic conditions [3,4,5,6]. In 2020, AMD was reported to affect approximately 196 million people worldwide and is expected to increase to affect 288 million people by 2040 [7]. Similarly, diabetic retinopathy was reported to affect 103.12 million adults worldwide and is expected to increase to affect 160.50 million people by 2045 [8]. These diseases (and others) unfortunately have the potential for devastating, and in some cases, irreversible vision loss. For example, the number of prevalent cases of low vision and blindness resulting from AMD was reported to be 8,057,521 globally in 2021 and is expected to increase to 13,880,610 by 2050 [9]. Additionally, the prevalence of clinically significant diabetic macular edema secondary to advanced stage diabetic retinopathy, is projected to increase from 18.83 million people in 2020 to 28.61 million people by 2045 [8].

Such retinal diseases often require frequent monitoring, timely intervention, and complex treatment to prevent irreversible vision loss [1,2]. Optical Coherence Tomography (OCT) has become a critical tool in the diagnosis and management of these conditions due to its ability to provide detailed imaging of retinal structures [2,10,11,12]. Given the high global retinal disease burden and the frequent monitoring and extensive therapies required to treat these diseases and their complications, the OCT technology is ideal for use in retinal disease due to its ability to capture images quickly and noninvasively [13,14,15]. However, despite its many advantages, accessibility to OCT remains a challenge, especially for elderly patients, those living in rural areas, and patients who require frequent follow-up appointments [16,17]. Traditional OCT systems require trained technicians and are therefore only available in clinical settings, further exacerbating the barriers to OCT-guided disease management, potentiating vision loss [18,19]. Recent advancements in biomedical engineering aim to address these limitations by introducing portable and community-based OCT solutions that improve accessibility. These improvements in accessibility include features such as technician-free use, remote interpretation, cost reduction, and deployment to community-based clinics.

## 2. Principles of OCT in Retinal Disease

New OCT form factors are emerging to address the growing need for accessible and patient-centered retinal care. The retina, a delicate layer of tissue at the back of the eye, is essential for vision and reflects systemic health [20]. For instance, hypertensive retinopathy reveals the effects of chronically elevated blood pressure, whereas diabetic retinopathy reflects microvascular damage from prolonged hyperglycemia [20]. Furthermore, the retina is the single tissue through which noninvasive imaging of blood vessels and neural tissue can be achieved, which contributes to the systemic insights the retina provides [21]. Early detection and continuous monitoring are crucial for effective management, yet access to diagnostic tools and interpretation of data for clinical care remains limited in many regions.

OCT has revolutionized retinal diagnostics by providing high-resolution, cross-sectional images of retinal structures. Since its introduction in the 1990s, OCT technology has progressed through substantial engineering advancements and has the gold standard test for screening, diagnosing, and guiding treatment decisions in retinal care. Compared to other imaging modalities of the retina, OCT offers several unique advantages. Fundus photography, while widely used, provides only a 2D surface view of the retina and lacks depth information [22]. Fluorescein angiography, while effective for visualizing vascular abnormalities, does not provide structural imaging and is more invasive than other imaging modalities, risking adverse effects such as allergic reaction to fluorescein [23]. Ultrasound B-scans, which are two-dimensional structural imaging “brightness scans”, are useful for imaging the posterior segment, such as visualizing retinal detachments, but have low resolution (~150 microns for a 10 MHz B-scan) [24].

Early OCTs, known as time-domain OCT (TD-OCT), used high-speed, continuous motion longitudinal scanning to increase the image acquisition rate, improving image quality and patient comfort, and transverse scanning to form two-dimensional images [13]. Following the inception of these early TD-OCT models, OCT technology has progressed through spectral domain OCT (SD-OCT), which has been further advanced by the development and clinical integration of swept-source OCT (SS-OCT) [14,15]. Features unique to TD-OCT, SD-OCT, and SS-OCT devices are outlined in Table 1. OCTs are often compared by their rate of obtaining A-scans, which are one-dimensional scans compiled to form cross-sectional images (B-scans) [25]. SS-OCT employs a tunable laser that rapidly sweeps across different wavelengths and provides faster imaging speeds (~100,000–236,000 A-scans/s) than previously available [15]. Its ability to simultaneously obtain information for all depth layers of the sample with higher sensitivity allows better penetration into deeper structures like the choroid, enabling detailed imaging of retinal and subretinal structures [26]. Major leaders in ophthalmic innovation, such as Heidelberg Engineering, Carl Zeiss Meditech, and Topcon Healthcare, have developed SS-OCT systems now standard in clinical practice. For example, Heidelberg Engineering’s Spectralis OCT incorporates eye tracking technology during image acquisition to enhance image reproducibility and reduce the duration of acquisition time [27,28]. Zeiss’s Cirrus OCT emphasizes a high axial scan acquisition rate and widefield visualization, facilitating detailed examination of peripheral retinal technologies [29,30]. Topcon’s Triton OCT provides multimodal imaging with integration of fundus photography and OCT angiography (OCTA), which is useful to visualize blood flow in the retina non-invasively, further expanding its utility [30,31].

**Table 1 bioengineering-12-00441-t001:** Summary of features unique to TD-OCT, SD-OCT, and SS-OCT [15,32].

Feature	TD-OCT (Time-Domain OCT)	SD-OCT (Spectral-Domain OCT)	SS-OCT (Swept-Source OCT)
Light Source	Broadband light source, split to sample and reference arms and interference detected by moving reference mirror	Broadband light with interference detected by spectrometer	Tunable laser swept across different wavelengths with interference detected by a single photodetector
Axial Resolution	8–10 µm	5–7 µm	11 µm
Scan Rate	400 A-scans/s	20,000–52,000 A-scans/s	100,000–236,000 A-scans/s
Clinical Utility	Basic imaging of retina	Standard for diagnosing and monitoring most retinal diseases	Choroid, anterior segment, and deep tissue imaging
Benefits	Lower cost	High-resolution, fast, widely available	Best depth penetration (able to obtain choroidal images), highly detailed
Limitations	Slow, low resolution, motion artifacts	Limited depth penetration	Costly, limited availability

Looking ahead, artificial intelligence (AI) is being increasingly integrated into OCT technology. By offering machine learning algorithms designed to automate image analysis, it has the potential to enhance clinical detection of retinal disease, particularly on a population level to aid in assessing large datasets [33]. Furthermore, AI can be employed to perform complex image processing tasks, such as segmentation and classification of retinal layers, vasculature, and pathologic features [34]. Segmentation models can distinguish the margins of retinal structures, allowing for precise quantification of retinal thickness and detection of abnormalities [34,35]. OCT-obtained retinal layer thickness can also be considered a useful biomarker in the future for predicting neurocognitive outcomes in pathologies like Alzheimer’s disease and Parkinson’s disease [36,37]. Moreover, classification models can be used to detect and grade diseases like AMD and diabetic retinopathy [38]. Deep learning models, including ResNet, have demonstrated incredible ability to distinguish between disease stages [38,39]. These AI-driven models not only improve diagnostic consistency but also reduce the time required for image interpretation, facilitating large-scale and remote screening initiatives and personalized treatment [38,40]. Recent advancements in deep learning have led to highly accurate models for detecting and segmenting retinal pathologies in real-world screening programs. Hybrid deep learning models combining convolutional and recurrent neural networks (CNN-RNN) have demonstrated robust performance in detecting diabetic macular edema (DME), achieving an area under the curve (AUC) of 0.94 in clinical settings [41]. Similarly, deep learning algorithms have shown high accuracy in segmenting key pathological features of neovascular age-related macular degeneration (nAMD), including intraretinal fluid (IRF), subretinal fluid (SRF), and neovascular pigment epithelium detachment (nPED) [34,35]. These models achieved AUC values ranging from 0.932 to 0.990 for IRF, 0.974 to 0.987 for SRF, and 0.961 to 0.969 for nPED [42]. Together, these advancements highlight the potential of deep learning in improving automated detection and segmentation of vision-threatening retinal diseases, enhancing early diagnosis and treatment strategies in ophthalmology.

Current OCT technology uniquely combines features of high resolution, non-invasiveness, and depth-resolved imaging, making it a highly efficient tool for screening for, diagnosing, and managing retinal diseases. It can detect microstructural changes in the retina before symptoms arise, such as subretinal fluid in AMD or DME [10]. Its ability to quantitatively track retinal thickness and changes in volume enables precise tracking of disease progression and response to treatment [2]. Additionally, because OCT is non-invasive and repeatable, it can be used frequently without causing patient discomfort [43]. OCT, for example, is especially critical in the management of diabetic retinopathy, where it identifies and monitors DME and guides treatment decisions (e.g., intravitreal anti-VEGF therapy) [44]. In AMD, OCT is essential for detecting the conversion from dry to wet AMD (a more severe form of AMD that causes rapid vision loss) and assessing the effectiveness of treatment [11]. In glaucoma, OCT measures retinal nerve fiber layer thickness, aiding in the early detection of optic nerve damage [12].

Despite its importance and effectiveness, the reliance on OCT for chronic disease management poses significant challenges. Current OCT models require technician special training to operate, requiring an office visit for each scan [18,19]. For conditions like AMD and diabetic retinopathy, patients often require frequent follow-ups, sometimes every 4–6 weeks, to monitor disease progression or response to treatment [45]. This places a substantial burden on patients and their families, particularly in terms of time, travel, and cost, especially for large healthcare systems covering widespread regional areas, such as the Veterans Health Administration. Elderly patients, who make up the majority of the affected population, often require transportation from caregivers due to their inability to drive after receiving a dilated exam [16,17]. Additionally, patients from rural or underserved areas may be disproportionately affected by these burdens since they may have limited access to OCT imaging, necessitating long travel times. Financial barriers also place limitations on OCT use, as OCT equipment is costly, making it less accessible in smaller clinics or community settings [19]. Health system limitations, such as overcrowded clinics and long appointment wait times, further hinder timely access to OCT, resulting in delayed diagnoses and treatment [46]. Without consistent follow-up, patients are at greater risk of disease progression, leading to irreversible vision loss and increased healthcare costs [47].

Current patient needs are being addressed by utilizing new tools built in interdisciplinary collaboration with biomedical engineering (BME). Portable, home-based OCT systems and AI-driven analysis presents the potential to bring imaging that is currently restricted to the ophthalmologic practice setting to community settings. This can mitigate barriers to eye care by enabling early detection and continuous monitoring without requiring frequent clinical visits. There remains, however, a pressing need to shift towards community-based retinal screening to address the growing prevalence of retinal diseases [48]. With recent treatment advances in retinal disease, early intervention is increasingly effective to preserve vision, making screening critical in the management of disease [49]. By bringing OCT into communities, it will reduce barriers to access, improve patient outcomes, and address the need for improved screening practices in retinal disease. The current landscape presents an opportunity to develop a novel OCT device that can meet these demands, transforming the future of retinal disease management.

## 3. Biomedical Engineering Innovations in OCT Technology

The development of handheld and portable OCT devices has significantly expanded the accessibility of retinal imaging. Notable among these innovations is the Envisu C2300 OCT by Leica Microsystems, which is FDA-cleared and weighs 1.5 kg, offering anterior and posterior segment imaging through various lenses [19]. Its lightweight design and flexible cable allow clinicians to use the device across diverse settings, from outpatient clinics to intensive care units. Similarly, Optovue Inc.’s iVue system, featuring a removable 2.2-kg scanner, facilitates rapid image acquisition, particularly beneficial for pediatric populations [50]. Heidelberg Engineering’s Spectralis Flex Module introduces a flexible boom arm that enhances adaptability in varied clinical environments [51]. Distinguishing features of these OCT devices are compared in Table 2.

Technological advancements have further propelled OCT capabilities through the integration of Vertical-Cavity Surface Emitting Laser (VCSEL) technology [52,53]. This innovation enables high-speed volumetric imaging, allowing real-time visualization of the retina and improving diagnostic accuracy. Additionally, the incorporation of Microelectromechanical Systems (MEMS) in OCT devices has led to significant reductions in size and cost while maintaining imaging quality comparable to conventional systems [52,54]. However, further refinement is required for these prototypes to achieve broader clinical adoption.

Smartphone integration in OCT technology offers another promising avenue for making retinal imaging more accessible. The OCTCHIP project exemplifies this trend, aiming to develop a compact, wireless OCT system. This approach was further advanced by Duke University, which created a low-cost OCT system using off-the-shelf components, totaling USD 7164. This cost-effective system could bridge the gap in access to retinal imaging in resource-limited settings [55]. Smartphone-based implementations, such as those developed by Subhash et al., digitize optical signals and display live OCT images, although real-time B-scan capabilities are still under development [56].

Home-based OCT solutions have emerged as a groundbreaking innovation in retinal care. Notal Vision’s Home OCT device, FDA-approved for self-operation, allows patients to capture images of their central macula at home. These images are analyzed by the Notal OCT Analyzer (NOA), an AI-driven platform that detects disease progression and informs personalized treatment strategies [57]. Similarly, the recently approved Scanly Home OCT system enables wireless transmission of SD-OCT images for AI analysis, effectively segmenting and quantifying hyporeflective spaces [57]. Additionally, proposed systems like SELF-OCT and Sparse OCT aim to provide low-cost home monitoring solutions for AMD, empowering patients to track disease progression without frequent clinical visits [58,59].

**Table 2 bioengineering-12-00441-t002:** Features of existing portable OCT devices.

Feature	Envisu C2300 OCT (Leica Microsystems)	iVue (Optovue)	Spectralis Flex Module (Heidelberg Engineering)	Notal Vision’s Scanly Home OCT
Size	Dimensions: NPD, handheld	Dimensions: 19.1 × 34.4 in (26.3–34.3 height), tabletop	Dimensions: NPD, tabletop (with flexible boom arm)	Dimensions: NPD, tabletop
Weight	3.31 lbs (1.5 kg)	79 lbs (35.83 kg)	NPD	NPD
Scan Rate	32,000 A-scans/s	80,000 A-scans/s	85,000 A-scans/s	10,000 A-scans/s
Axial Resolution	4 µm	5 µm	6 µm	19 µm
No Technician Required	No	No	No	Yes
Deployment	Clinic	Clinic	Clinic	Home
Clinical Significance	Diagnosis, disease monitoring	Diagnosis, disease monitoring	Diagnosis, disease monitoring	Screening
Cost	NPD	$10k for base model	NPD	NPD

NPD: not publicly disclosed [19,50,51,57,60].

## 4. SightSync: A Community-Model OCT

Accessibility remains one of the most prominent systems challenges for retinal monitoring. Home monitoring systems offer convenience for close monitoring in outpatient settings, however, the economics of placing a >USD 10,000 medical device in every patient’s home who requires close monitoring is not feasible [19,57]. Therefore, our group has developed a “community-based” OCT for serving multiple patients within a single geographic area with a single device. This device is aimed at (1) providing an easy-to-use system that can be accessed by multiple patients without the need for a technician and (2) a method to ensure data security in a publicly accessible medical device within an outpatient or community setting [59,61].

## 5. Technicianless, Easy-to-Use OCT

Outside of the Notal Vision ‘Home Optical Coherence Tomography (OCT)’, nearly every clinical OCT is designed for image acquisition with the aid of a trained technician. Mainly, these technicians are responsible for patient orientation, image quality assurance, and image volume captures. The American Telemedicine Association’s Practice Guidelines for Ocular Telehealth-Diabetic Retinopathy emphasize that imaging personnel must possess the knowledge and skills to independently acquire retinal images, ensuring high-quality imaging even in the absence of a licensed eye care professional [62]. Additionally, the review by Chopra et al. highlights that the operation of clinical OCT systems typically requires skilled personnel due to the complexity and precision needed for high-quality image acquisition [19]. Therefore, a device that is patient-operated without a technician will need to both orient the patient properly and acquire images of sufficient quality.

The SightSync device ensures precise patient orientation through a combination of mechanical and programmatic features, aligning the eye at the sensor midline and accurately targeting the retina’s focal depth. To achieve this, we integrated the Lumedica OQ LabScope 3.0 XR, an affordable, off-the-shelf spectral-domain optical coherence tomography (SD-OCT) system, into our design [63,64]. We implemented specific hardware interface modifications and developed a software data transfer wrapper to tailor the system to our requirements.

The Lumedica OQ LabScope 3.0 XR operates at a center wavelength of 850 nm, providing an axial resolution of 5 µm in air (3 µm in tissue) and a transverse resolution of 18 µm. It offers an axial scan range of 2 mm in air (1.5 mm in tissue) and a transverse (linear) scan range of 7 mm, with a volume scan range of 5 mm × 5 mm. The system achieves an A-scan line rate of 80 kHz and a B-scan image rate of 50 scans per second, capturing images at a resolution of 512 × 512 pixels. Despite its advanced capabilities, the OQ LabScope 3.0 XR maintains a compact form factor, with system dimensions of 19 cm × 33 cm × 15 cm and a weight of 2.7 kg.

In the SightSync device, patient orientation is ensured by both mechanical and programmatic features to ensure that the eye is at sensor midline and that the focal length depth is correctly targeting the retina. The eyepiece, which is built around Lumedica’s sensor arm, contains a spring-loaded mechanism that allows for manual depth adjustment (Figure 1). Additionally, the foam exterior blocks external light from reaching the sensor, which prevents signal quality reduction. Focal length sweep using a liquid lens is used in the SightSync for dynamic depth adjustment during scan acquisition. Liquid lenses, which utilize an adjustable refractive index through fluid manipulation, enable rapid changes in focal length without the need for mechanical movement of traditional optical elements. By performing a focal length sweep, the system can acquire images at various depths, enhancing the ability to isolate the target plane and improve depth of field. This approach is particularly effective to find the proper plane and maximize the signal-to-noise ratio (SNR) [65].

Device form factor plays a critical role in its successful deployment in public settings, as it directly influences usability, space efficiency, and adaptability to diverse environments. A compact and versatile design ensures the device can seamlessly integrate into various public spaces without disrupting the existing infrastructure. Incorporating a dynamic arm with three degrees of freedom significantly enhances the device’s flexibility, enabling deployment in multiple configurations such as table-top setups, wall mounts, or walk-up stations. This adaptability allows the device to cater to different user needs and spatial constraints while maintaining optimal functionality. Whether installed on a crowded counter, mounted on a vertical surface to save floor space, or positioned for easy walk-up access, the device remains accessible and ergonomic, ensuring a smooth and user-friendly experience across various public applications. 

## 6. Secure Data Transfer in a Public Setting

In the context of publicly accessible medical devices, such as kiosks or mobile clinics, the secure handling of patient data is not only a best practice—it is a legal and regulatory necessity. In the United States, the Health Insurance Portability and Accountability Act (HIPAA) mandates strict requirements for the protection of electronic protected health information (ePHI). In the European Union, the General Data Protection Regulation (GDPR) enforces equally stringent rules for the collection, processing, and storage of personal health data (PHD).

While the transmission of lightweight numeric vitals (e.g., heart rate or blood pressure) via low-bandwidth protocols such as Bluetooth, LoRaWAN, or Wi-Fi is feasible and common, the transfer of high-resolution data, such as full-volume Optical Coherence Tomography (OCT) scans, presents both bandwidth and security challenges [66]. For instance, LoRaWAN, while suitable for long-distance communication, offers transmission speeds between 0.3 kbps and 50 kbps, which are insufficient for large medical image files [67]. Similarly, Bluetooth Low Energy, with a bandwidth of 1 Mbps and a range of 15 to 30 m, may not support the rapid transfer of high-resolution OCT data [68]. Furthermore, maintaining a persistent connection to public Wi-Fi in settings like outpatient clinics or pharmacies exposes the system to increased risks of unauthorized access or data breaches [69,70].

The SightSync OCT system is designed to support secure, real-time data transfer in low-connectivity environments through a privacy-by-design architecture (Figure 2). It leverages an embedded MQTT (Message Queuing Telemetry Transport) broker and acts as a self-hosted Wi-Fi access point (AP). This allows the OCT device to wirelessly connect to a patient’s smartphone or a clinician’s tablet without relying on external internet access.

Once connected, the MQTT broker transmits real-time retinal scan data, including high-resolution images and diagnostic insights, to an MQTT client on the mobile device. This direct, low-latency communication supports instantaneous viewing and analysis of results, even in remote settings or underserved areas. If internet becomes available later, the mobile device can securely sync stored OCT data to a HIPAA-compliant cloud platform or hospital EHR system, enabling remote specialist review and long-term monitoring.

This architecture enhances accessibility and operational flexibility, while ensuring that sensitive patient data remains protected at all times. To maintain HIPAA compliance, the system addresses all three regulatory safeguard domains: administrative safeguards, technical safeguards, and physical safeguards. 

To ensure compliance with HIPAA’s administrative safeguards, the SightSync OCT system incorporates role-based access control, allowing only authenticated users, such as patients and clinicians, to access scan data via the mobile application. Each session is protected through secure login mechanisms, with the option to implement multi-factor authentication (MFA) for added security. Additionally, the embedded MQTT broker generates comprehensive audit logs, recording data transactions with timestamps, session metadata, and user or device identifiers. These logs support real-time monitoring, accountability, and breach traceability.

From a technical standpoint, the system safeguards data using end-to-end encryption with AES-256, protecting patient information both during transmission and at rest [71]. All MQTT communication runs over Transport Layer Security (TLS), which prevents man-in-the-middle attacks and ensures that data remains private [72]. The system also includes data integrity checks, verifying the completeness and accuracy of transmitted scans. When the BYOD (Bring Your Own Device) client later connects to the internet, it can securely sync OCT data to a HIPAA-compliant cloud platform that aligns with interoperability standards such as HL7 and FHIR, enabling downstream clinical integration. In addition, this architecture achieves an effective throughput of approximately 16–64 Mbps. The lightweight nature of the MQTT protocol, combined with minimal overhead from TLS—typically introducing only 100–200 ms of latency during the initial handshake—allows a fully processed OCT volume scan (15–25 MB) to be securely uploaded in under 10 s per scan.

Lastly, the system adheres to HIPAA’s physical safeguards by minimizing data storage on the device itself. The embedded hardware, whether a Raspberry Pi or similar microcontroller, only buffers or temporarily stores scan data, significantly reducing exposure in the event of theft or physical compromise. Furthermore, the device is hardened at the firmware and OS level, with unnecessary services disabled and remote access tightly restricted. The SightSync OCT creates a closed, self-hosted Wi-Fi network, eliminating reliance on public infrastructure and reducing susceptibility to unauthorized external connections. This isolated design ensures patient data remains secure, even in decentralized or public-facing deployment environments.

## 7. Image Quality Validation and Analysis

A publicly accessible, patient-operated OCT device requires advanced image analysis to ensure scan quality, detect emergencies, and provide quantitative retinal health metrics. The workflow begins with an image quality assessment, employing deep learning classifiers and traditional image processing metrics such as signal-to-noise ratio (SNR), contrast-to-noise ratio (CNR), edge detection (Sobel, Canny), and blur detection (Laplacian variance) to determine scan interpretability [73,74]. If the image quality is inadequate, the system prompts the user to retake the scan with guidance on improving positioning and focus. Preliminary user obtained images from the SightSync (N = 40 images) had a CNR of 2.08 ± 1.05 and a SNR of 1.21 ± 0.19, which is comparable to Lumedica’s technician-obtained OCT images (CNR = 1.592 ± 0.021) and that of the Heidelberg Spectralis (CNR = 1.687 ± 0.027) (Figure 3) [64]. It is worth noting that while user-obtained images from SightSync had higher variability in signal quality compared to clinically obtained OCT images, it was still capable of obtaining sufficient signal without technician aid for alignment (Figure 4).

Following quality verification, the device conducts emergency detection through machine learning-based classification models trained to identify critical conditions such as retinal detachment, macular edema, and vitreous hemorrhage. These models utilize convolutional neural networks (CNNs) and anomaly detection techniques like autoencoders to recognize deviations from normal scans (outlined in Table 3) [75,76]. If a high-confidence emergency is detected, the system alerts the user and advises immediate medical attention while offering telemedicine integration for expert review [77].

**Table 3 bioengineering-12-00441-t003:** Machine learning algorithms used for OCT image pathology detection. Note that this list is not exhaustive, but includes major models developed over the past seven years.

Model (Year)	Pathologies Detected	Performance (Metrics)	Source (Reference)
Kermany et al. CNN (2018)	Choroidal neovascularization (wet AMD), diabetic macular edema (DME), drusen (dry AMD), and normal retina.	Accuracy ≈ 96.6%, Sensitivity 97.8%, Specificity 97.4%, AUC 0.999 in classifying OCT scans (AMD/DME vs. others)	Cell (2018)—UCSD/Mendeley OCT dataset [78]
DeepMind OCT AI (2018)	Over 50 retinal conditions (e.g., age-related macular degeneration, diabetic eye disease, retinal detachment, etc.)	AUC > 0.99 for most conditions (≥0.96 for all); ~94% accuracy in recommending correct referral urgency	Nature Medicine (2018)—Moorfields Eye Hosp. & DeepMind [79]
Moorfields/DeepMind AMD Prognosis (2020)	Risk of conversion to exudative (“wet”) AMD in patients with early/intermediate AMD	Performed as well as or better than expert clinicians in predicting 6-month progression to wet AMD (higher predictive accuracy)	Nature Medicine (2020)—Moorfields Eye Hosp. & DeepMind [80]
f-AnoGAN (2019)	Unsupervised anomaly detection—tested on OCT identifying retinal fluid anomalies (e.g., fluid in AMD or DME).	AUC ~0.85 and Sensitivity ~88% for detecting anomalous OCT B-scans with fluid (outperformed other methods in experiments)	Med. Image Anal. (2019)—Schlegl et al. (Medical Univ. Vienna) [81]
3D ResNet (CUHK) Glaucoma AI (2019)	Glaucomatous optic neuropathy (detection of glaucoma from optic-disc OCT volumes)	Primary validation AUC 0.969, Sensitivity 89%, Specificity 96%, Accuracy 91% (similar performance on external test sets).	Lancet Digital Health (2019)—Li et al. (CUHK/Stanford) [82]
RETFound Foundation Model (2023)	Multiple retinal diseases (foundation model pretrained on OCTs, fine-tuned for AMD, diabetic retinopathy, etc.)	Achieved state-of-the-art on various eye disease detection tasks, outperforming conventional models with fewer labels	Nature (2023)—Zhou et al. (Moorfields/UCL) [83]
AMD Stage Classifier—PINNACLE (2023)	Age-related macular degeneration stages: normal, intermediate (iAMD), geographic atrophy (GA), neovascular AMD (nAMD)	ROC–AUC ≈ 0.94 (averaged over AMD stage classifications) on real-world OCT volumes; balanced accuracy ~90% on internal test	Sci. Reports (2023)—Leingang et al. (Med. Univ. Vienna) [84]
ZEISS CIRRUS PathFinder (2024)	Macular OCT abnormalities (flags B-scans with subretinal fluid, intraretinal fluid, RPE atrophy or elevation, retinal layer disruptions, etc.)	~88% sensitivity and 93% specificity for automatically detecting OCT scans with pathological findings (internal validation).	ZEISS CIRRUS PathFinder (2024)—FDA-pending product [85]

For long-term retinal health monitoring, image segmentation and quantitative analysis extract key retinal features and measure layer thicknesses over time. Advanced segmentation techniques, including deep learning models like U-Net and DeepLabV3, graph-based algorithms, and active contour methods, delineate structures such as the retinal nerve fiber layer (RNFL), ganglion cell layer (GCL), and retinal pigment epithelium (RPE) (Table 4) [39,86]. These measurements enable tracking of disease progression, such as RNFL thinning in glaucoma or retinal swelling in diabetic macular edema, by generating trend graphs, volumetric assessments, and longitudinal comparisons.

**Table 4 bioengineering-12-00441-t004:** Notable machine learning algorithms used for OCT image segmentation over the past 7 years.

Model (Year)	Segmented Structures/Pathologies	Performance Metrics	Source (Reference)
U-Net (2015) Applied ~2017	Retinal layers; lesions (e.g., fluid)—widely used baseline	High accuracy on healthy layers; degraded on complex pathology (lower IoUs)	Ronneberger et al., MICCAI 2015, [87]
ReLayNet (2017)	Seven retinal layers + intraretinal fluid (DME)	DSC ~0.77 for fluid (vs 0.58 human); ~0.9 for layers—outperforms prior methods (fluid DSC 0.28–0.67)	Roy et al., Biomed. Opt. Express 2017, [88]
RNN + Graph Search (RNN-GS, 2018)	Retinal layer boundaries (7 in normal; 3 in AMD eyes)	Mean boundary error ~0.53 px (normal) and 1.17 px (AMD)—competitive with CNN-based method	Kugelman et al., Biomed. Opt. Express 2018, [89]
DeepMind 3D U-Net (2018)	15-class tissue map: retinal layers, fluids (IRF/SRF), PED, etc.	Enabled expert-level diagnosis (AUC ≈ 99%) using segmented maps; device-agnostic tissue representation	De Fauw et al., Nature Medicine 2018, [90]
DeepLabV3 (2018)	Retinal layer surfaces (e.g., ILM, RPE) in AMD patients	Low boundary errors (comparable to U-Net); slightly better RPE segmentation on Spectralis OCT	Devalla et al., Ophthalmology Science 2023, [91]
DME-DeepLabV3+ (2023)	Diabetic Macular Edema (fluid regions)	MIoU ≈ 91.2%, F1 ≈ 91.2%, Pixel Acc. 98.7% on DME vs. background	Guo et al., Frontiers 2023, [92]
Google “Whole-Volume” (2021)	Pathologies in AMD/DME: IRF, SRF, sub-RPE material, PED	Dice 0.43–0.78 (varies by lesion type; e.g., best for large fluid),—rated equal to or better than one expert in 73% of scans	Wilson et al., JAMA Ophthalmology 2021, [93]
GANSeg (2023)	Seven retinal layers + intraretinal fluid (cross-device adaptation)	Layer Dice up to 90% (GCL + IPL); IRF Dice ~58% (vs 79% human)—generalized to unseen device data	Lee et al., Ophthalmology 2023, [94]
DeepLabV3+ w/CPS (2024)	10 retinal layers + 4 features (cysts, collapsed layers, etc. in MacTel)	Semi-supervised approach achieved highest IoUs on all 14 classes—significantly outperforms supervised U-Net, ReLayNet, etc.	Shi et al., TVST 2024, [95]
RetinAI Discovery (2023) (Industry)	Seven retinal layers (RNFL through RPE), choroid, subretinal & intraretinal fluid, sub-RPE material, hyper-reflective foci, etc.	N/A (Research-use only; “reliable” automated detection reported)	RetinAI (Company release) 2023, [96]

A publicly available OCT device will also significantly enhance the availability of scan libraries suitable for training machine learning algorithms. By collecting anonymized scans from a diverse patient population, the system can contribute to larger, more representative datasets that improve model accuracy and generalizability. This expanded dataset allows for better refinement of deep learning models, leading to improved detection and segmentation performance across different demographic groups and disease stages. Additionally, federated learning can enable continuous updates to the models while maintaining user privacy, ensuring that new pathological patterns are recognized more effectively over time.

## 8. Conclusions

The growing prevalence of retinal disease demands rigorous monitoring to inform timely intervention, however, traditional OCT systems pose barriers to accessibility that limit its efficacy. Solutions such as the SightSync OCT demonstrate how community-based, technicianless OCT models can alleviate logistical and financial barriers while maintaining high-quality imaging and secure data management. The device’s capabilities, including patient self-administration, variable-depth image acquisition, real-time data transfer, and a compact, adaptable design, may contribute to improved accessibility in retinal imaging. The current SightSync model, for which patent protection has been secured, is undergoing clinical validation in an IRB-approved study at the Louis Stokes Cleveland VA Medical Center. As with other accessibility-focused screening and diagnostic technologies, similar devices intended for deployment in community or non-traditional clinical settings will be required to meet regulatory standards under the FDA’s de novo classification process and demonstrate compliance with HIPAA regulations for secure data handling and transmission. Given its portability and potential for remote analysis, the SightSync may be well-suited for integration into ophthalmology and optometry practices, community health clinics, and resource-limited environments.

As biomedical engineering continues to drive advancements in OCT technology, integrating patient-centered, community-based innovations into routine clinical practice presents an opportunity to greatly reduce the burden that retinal diseases place on patients, caregivers, and the healthcare system. By embracing these developments, providers and institutions can progress toward a more proactive, accessible, and efficient approach to retinal care and the prevention of vision loss.

## Figures and Tables

**Figure 1 bioengineering-12-00441-f001:**
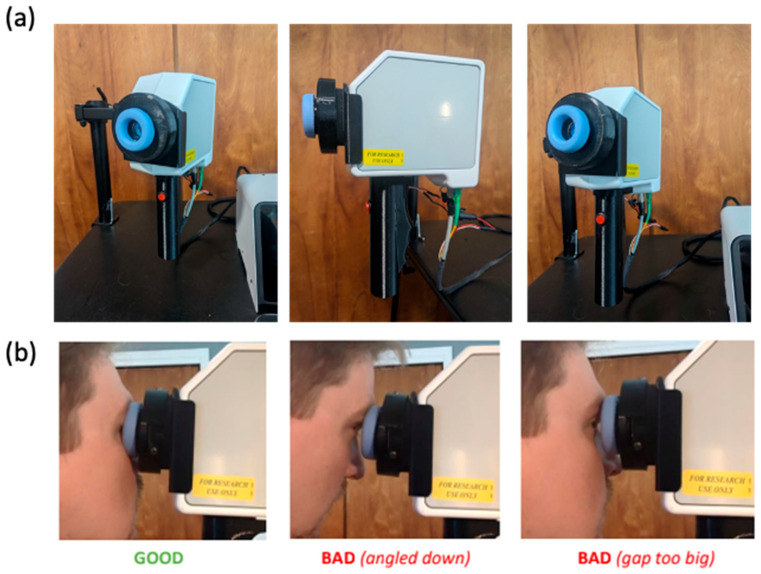
(**a**) Physical interface of SightSync, including a dynamic arm with three degrees of freedom to adjust for height and tilt. Mobile arm containing optics connected to a central housing with spectrometer and other OCT hardware. (**b**) Physical orientation features of the SightSync device include a displaceable foam padding which allows users to adjust their depth while blocking external light from introducing noise to the sensor.

**Figure 2 bioengineering-12-00441-f002:**
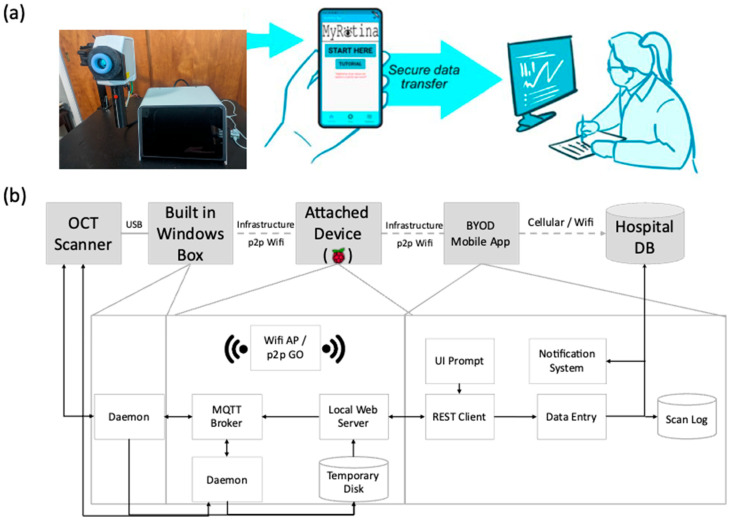
(**a**) Conceptual rendering of an OCT (Optical Coherence Tomography) scanner integrated with a robotic arm for retinal imaging. This setup is designed to enhance accessibility and precision in retinal diagnostics. (**b**) A system architecture diagram illustrating the workflow of a secure retinal imaging and data transfer system. The OCT scanner captures retinal images, which are processed by a built-in Windows-based system. A Raspberry Pi-based attached device facilitates wireless data transmission to a BYOD (Bring Your Own Device) mobile application, which then securely uploads the data to the hospital database for clinical review and analysis. The system integrates MQTT for communication, a REST client for data handling, and a notification system for updates.

**Figure 3 bioengineering-12-00441-f003:**
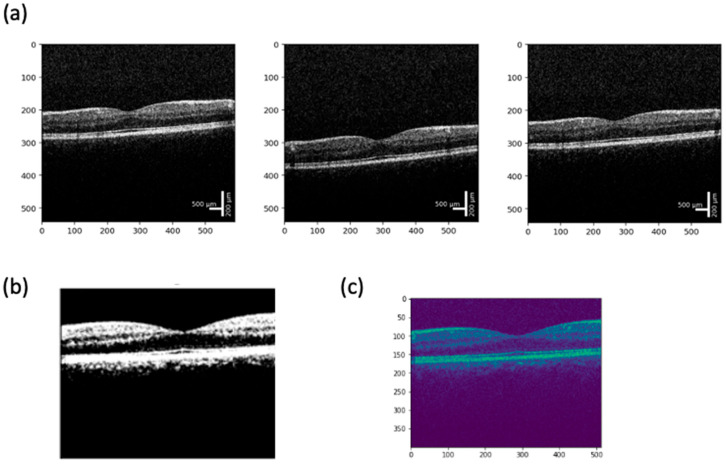
(**a**) Sample OCTs obtained from the SightSync prototype from three users who self-administered their scans. X– and Y–axes are in pixels, scale bars are approximated from device technical specifications; (**b**) thresholded OCT scan image of self-administered OCT, used for quality determination; and (**c**) layer segmentation using open-source CIDAS CLIPSeg library (2022) in Python (2022).

**Figure 4 bioengineering-12-00441-f004:**
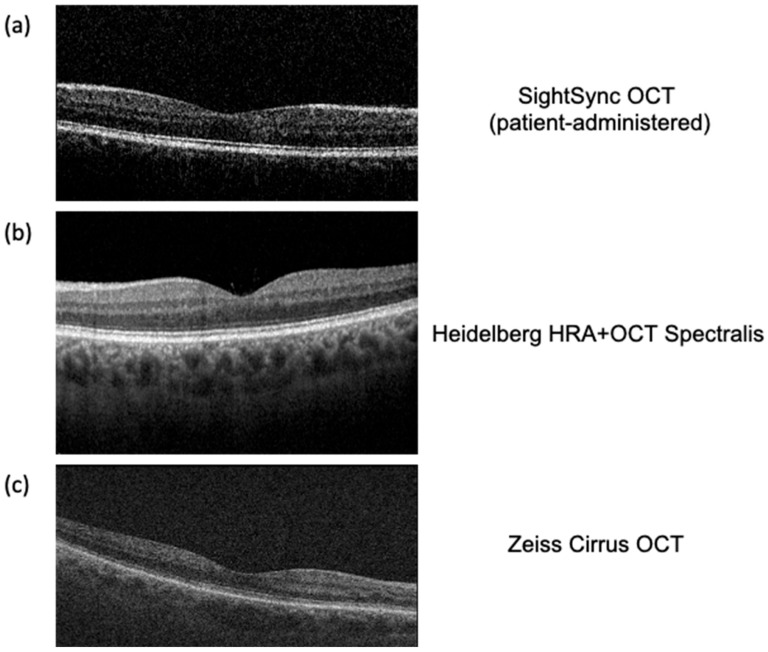
Comparison OCT images taken by (**a**) SightSync (patient-administered); (**b**) Heidelberg HRA + OCT Spectralis (technician administered); and (**c**) ZEISS CIRRUS OCT (technician administered).

## Data Availability

The data presented in this study are available on reasonable request from the corresponding authors. Availability is subject to the discretion of the authors.

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
