# Peer review of "Recent Optical Coherence Tomography (OCT) Innovations for Increased Accessibility and Remote Surveillance"

_bioengineering, 2025, doi:10.3390/bioengineering12050441_

Round 1
Reviewer 1 Report
Comments and Suggestions for Authors
The paper by Devine et al. describes a community-based, technician-free OCT device designed to enhance accessibility while ensuring secure data transfer and high-quality imaging. The authors described the history of OCT implementation in retinal disease analysis and briefly presented a novel OCT device. The theme of the study is important and interesting to a wide range of readers; however, prior to publication, several issues should be addressed:
-
The introduction looks really tiny; please add more general information about OCT implementation in ophthalmology.
-
Illustrations and photographs will help a lot to evaluate the progress of OCT devices (in sections 2 and 3).
-
Section 4 is limited. No references, no information about the description of utilized components, and no information about device construction.
-
As I understand, Section 7 is missing exact values about different image qualities. The authors propose to judge the quality of OCT images only based on visual examination. I believe that the authors should provide some impersonal criteria for the evaluation of image quality.
-
Presented images are missing axes labels and dimensions.
-
It will help the readers to understand the peculiarities of the proposed device if the authors describe them more precisely in the abstract and in the conclusions.
The paper may be published after correction of the mentioned issues.
Author Response
The paper by Devine et al. describes a community-based, technician-free OCT device designed to enhance accessibility while ensuring secure data transfer and high-quality imaging. The authors described the history of OCT implementation in retinal disease analysis and briefly presented a novel OCT device. The theme of the study is important and interesting to a wide range of readers; however, prior to publication, several issues should be addressed:
- The introduction looks really tiny; please add more general information about OCT implementation in ophthalmology.
- We appreciate the reviewer’s suggestion. In response, we have expanded the introduction to include a more comprehensive overview of OCT use in ophthalmology, while still valuing conciseness in the information presented. We have also expanded the introduction to discuss the global disease burden of the conditions which can be monitored by OCT technology. The revised introduction now provides greater context for the innovations discussed in the rest of the manuscript.
- Illustrations and photographs will help a lot to evaluate the progress of OCT devices (in sections 2 and 3).
- We appreciate the reviewer’s suggestion. In lieu of photographs of OCT devices as they have progressed, we have included a table that compares the key features of these devices. We believe this will help the audience to draw direct comparisons between the devices using metrics that are discussed throughout the paper, drawing attention to their specific benefits and limitations.
- Section 4 is limited. No references, no information about the description of utilized components, and no information about device construction.
We have rewritten this section to provide more information and added references for where we sourced our hardware. If readers are interested in specific mechanics of the hardware, they can look at the referenced papers from Lumedica, but we outlined the key technical specifications of the device.
- As I understand, Section 7 is missing exact values about different image qualities. The authors propose to judge the quality of OCT images only based on visual examination. I believe that the authors should provide some impersonal criteria for the evaluation of image quality.
We agree with the reviewer that metrics such as image quality are essential for readers to determine the differences between technician and technicianless OCT scans. While SightSync is not yet clinically tested (IRB is approved and recruiting soon at Case Western Reserve University) we did include some preliminary analysis on images taken by demo users and added in comparable metrics of other systems, including our hardware (Lumedica EyeScope) in a technician acquired trial. We hope this sufficiently adds context for the reader, however, we still believe providing exemplar images for competitive systems is appropriate for readers to compare imaging modalities qualitatively.
- Presented images are missing axes labels and dimensions.
Axes were added to these images for clarity and estimated scale bars were added based on technical specifications of the hardware used.
- It will help the readers to understand the peculiarities of the proposed device if the authors describe them more precisely in the abstract and in the conclusions.
- We thank the reviewer for their suggestion. We have modified the abstract and conclusion to provide an overview of the specifications and benefits of the SightSync device.
The paper may be published after correction of the mentioned issues.
Reviewer 2 Report
Comments and Suggestions for Authors
This manuscript presents a timely and highly relevant review of biomedical engineering advances in OCT, focusing on accessibility, portability, and technician-free models such as SightSync. The scope aligns well with the goals of Bioengineering. The article is well-structured and clearly written, covering key technological developments in OCT, including home-use systems, MEMS and VCSEL technology, AI integration, and secure data transfer.
However, the manuscript lacks deeper technical benchmarking between systems and over-relies on descriptive text rather than comparative analysis. Additionally, key recent works on OCT miniaturization, smartphone OCT, and regulatory standards are not covered. Image quality validation methods are not thoroughly discussed or referenced with primary data.
The manuscript should undergo major revision before it can be considered for publication
- Abstract: Current version effectively summarizes the key goals but should include more quantitative indicators, like diagnostic accuracy statistics for context
- Introduction:
- Authors are advised to Add recent statistics on global retinal disease burden (e.g., WHO/GBD 2024).
- Consider defining “accessibility” with concrete benchmarks—technician-free use, remote interpretation, or cost reduction.
- Principles of OCT in Retinal Disease
- It is suggested that authors include a summary table comparing different OCT systems in terms of axial resolution, scan rate, and clinical utility, pros and cons of different systems, etc,
- Terms like “A-scan” and “B-scan” should be briefly explained for non-specialist readers.
- Biomedical Engineering Innovations in OCT Technology
- A comparative table needs to be created summarizing device size, weight, imaging specifications, and target use cases.
- Include cost comparison, where available, and practical limitations in deployment.
- SightSync: A Community-Model OCT
- Key performance claims should be benchmarked against commercial OCT systems (e.g., resolution, scan repeatability, false positive/negative rates). Add quantitative validation of image quality using accepted OCT metrics.
- Secure data transfer in a public setting
- Include discussions on HIPAA and GDPR compliance.
- adding a performance benchmark (latency, data throughput) will add value to manuscript.
- Image Quality and AI Analysis
- List the datasets used (e.g., EyePACS, RETOUCH, ORIGA), performance metrics (AUC, sensitivity/specificity), and test sizes.
- The segmentation tools (U-Net, CIDAS, DeepLabV3) are mentioned but could benefit from performance summaries.
- Conclusion: It is highly recommended to include a brief roadmap for real-world deployment, regulatory clearance, and market translation.
Author Response
We agree with the reviewer that conducting a literature review on machine learning models would add value to the review paper overall. We have accordingly added 2 tables outlining prevalent machine learning pathology detection models and segmentation models along with performance metrics when available.

Round 2
Reviewer 1 Report
Comments and Suggestions for Authors
The authors addressed arised issues; the paper may be published.
Reviewer 2 Report
Comments and Suggestions for Authors
The authors have addressed all comments raised during the previous review round.
On a separate note, authors are advised to include the line numbers and pages in their cover letter, clearly indicating where the changes/additional information were included in the revised manuscript, and to use yellow or green highlight to the changes/added new information to facilitate the manuscript assessment.